# Creating interventions to transition long-lasting insecticide net distribution in Ghana

Franklin Glozah,[1] Emmanuel Asampong ![ORCID],[1] Philip Teg-Nefaah Tabong,[1] Adanna Nwameme,[1] Ruby Hornuvo,[1] Margaret Chandi,[2] Nana Yaw Peprah,[2] Philip Adongo,[1] Phyllis Dako-Gyeke ![ORCID] [1]

[1]Social and Behavioural Sciences, University of Ghana School of Public Health, Legon, Accra, Greater Accra, Ghana
[2]Ghana Health Service, Accra, Greater Accra, Ghana

**Correspondence to**
Dr Emmanuel Asampong;
easampong@ug.edu.gh

## ABSTRACT

**Objective** Mass long-lasting insecticide net (LLIN) distribution campaigns are rolled out, as a part of the Ghana Malaria Strategic plan (2021–2025) which seeks to protect at least 80% of the population at risk with effective malaria prevention interventions. Although the mass LLIN distribution campaign indicates a comprehensive stakeholder engagement approach, it does not systematically transition into the basic primary healthcare structures within the Ghana Health Services. This paper presents the process and outcome of creating an innovative social intervention, which focuses on community mobilisation and capacity building of community health officers.

**Methods** This study employed a concurrent triangulation mixed methods approach conducted across six districts in Eastern and Volta regions, Ghana. Findings were synthesised, grouped and further distilled to guide the participatory cocreation workshops. Cocreation involved participatory learning in action technique which is a practical, adaptive research strategy which enabled diverse groups and individuals to learn, work and act together in a cooperative manner.

**Results** The results suggest the establishment of a Community Health Advocacy Team (CHAT). This would be necessary in efforts aimed at transitioning LLIN distribution campaign in communities. The role of the CHAT would be centred on key elements of community/social mobilisation and capacity building, all nested in a social and behaviour change communication strategies.

**Conclusion** The research team is in the process of assessing the acceptability and feasibility of the CHAT intervention with all stakeholders in the various communities. Assessment of the effectiveness of the CHAT intervention would be done at a later time.

## INTRODUCTION

Malaria continues to be endemic in most parts of Africa, including Ghana, causing millions of lives to be lost across age categories.[1 2] There was however progress between 2016 and 2019 where Ghana made significant progress in malaria control. Cases decreased by 32% (from 237 cases per 1000 of the population at risk to 161 cases), and deaths decreased 7% (from 0.4 per 1000 of the population at

**STRENGTHS AND LIMITATIONS OF THIS STUDY**

⇒ Periodically, the National Malaria Control Programme rolls out mass long-lasting insecticide net (LLIN) distribution campaigns in communities. However, it has not been embedded into the primary healthcare structures. This study seeks to cocreate an intervention that hinges on community mobilisation and community health officers playing key roles in the distribution campaigns.
⇒ This study suggests a community health advocacy team to provide support services to transition LLIN distribution campaign within communities.
⇒ LLIN distribution campaign would be transitioned into the community-based health planning and services, improve use of LLIN within communities, thereby reduce malaria morbidity and mortality.

risk to 0.37).[3] Mass long-lasting insecticide net (LLIN) distribution campaigns are rolled out, as a part of the Ghana Malaria Strategic Plan (2021–2025) which seeks to protect at least 80% of the population at risk with effective malaria prevention interventions. In spite of progress made in overall LLIN ownership, challenges still remain. The Ghana Malaria Indicator Survey shows 67% access to a LLIN, with 43% use.[4] The inference therefore is that people have not taken up LLIN despite the distribution campaign. Although these campaigns expose high proportions of the Ghanaian community to LLIN interventions, they hardly lead to desired health-related behaviours (ie, LLIN use).

Some barriers to LLIN use have been documented to include distribution challenges where there is inadequate number of nets per household, limited social and behaviour change communication (SBCC) activities as well as lack of continuous malaria education.[5–8] At the community level, there is knowledge gap on malaria prevention, inability to hang LLINs in many households due to housing type and

sleeping places. There is also misuse and repurposing of LLINs.[9 10] Health-worker challenges have also been reported to include lack of adequate training on community mobilisation skills, minimal number of staff and lack of follow-up, community engagement, and supervision.[11 12]

Thus, ability to achieve desired outcomes from LLIN campaigns may require adoption of social innovative approaches which support behaviour change within communities.[13]

Social innovation is described as a collective process enabling the generation of ideas by people who participate collaboratively to improve delivery strategies in the community or health facility.[14] The social objective emphasises the engagement of concerned communities within which interventions will be diffused with innovative approaches meeting both social and medical needs.[14] Community directed programmes provide opportunities for government, health services, social actors and individuals to work closely with populations directly affected by diseases, especially infectious conditions. One such example is the Community-based Health Planning and Services (CHPS) program in Ghana, which seeks to ensure accessible, equitable, efficient and good quality healthcare services.[15] The CHPS concept involves the provision of door-to-door primary healthcare services to community members by trained nurses known as community health officers (CHOs).[15–17] CHOs provide antenatal care, family planning, health education, outreach clinics for delivery of child welfare services, and school health services.

In order to create a people centred intervention, the project (Improving the Effectiveness of Mass Long Lasting Insecticide-treated Net Distribution Campaigns Through Community-based Health Planning and Services Programme in Ghana) seeks to leverage on the CHPS to ensure community involvement and ownership of the LLIN Mass Distribution Campaigns in Ghana. Although some community health workers (eg, health volunteers, community health nurses) participate in the registration of households, list potential beneficiaries, and distribute LLINs during the campaigns, their involvement is paid for on a contractual basis. To ensure sustainability, the need to transition this campaign into CHPS has become imperative as the National Malaria Control Programme (NMCP) continues to rollout the 2021 campaign, amidst the COVID-19 pandemic. This paper presents the process and outcome of the cocreation of an innovative social intervention which would transition the Mass LLIN distribution campaign into CHPS by focusing on community mobilisation and capacity building.

## METHODS
### Study site
A total of six communities, one community per district across two regions in southern Ghana, participated in the study. These were communities in districts where the 2021 point mass distribution (PMD) campaigns of LLINs are ongoing. PMD is one of the strategies adopted by the Ghana Health Service (GHS) and its partners' whereby only designated sites are noted for distribution of LLINs. This was to avoid possible biases regarding community engagement by ensuring that components align with the timelines of the NMCP and the funder. The study was conducted in districts with the highest prevalence of malaria as reported in the District Health Information Management System (DHIMS2), a comprehensive web-based application for remotely compiling data across different levels of a health system into a central storage point: Ho West (Tsito-90%), Ho (Takla Hokpeta-75%) and Agortime Ziope (Kpetoe-100%) in the Volta Region; and Birim South (Apoli-94%), Achiase (Achiase-94%) and Abuakwa North (Kukurantumi-93%) in the Eastern Region (data source: DHIMS 2).

### Patient and public involvement
Participants in this study were not involved in setting the research question or the outcome measures. However, they were closely involved in the creation of the Community Health Advocacy Teams (CHAT). Participants were also involved during inauguration of CHAT which helped to motivate community involvement during the study.

### Population and sample
The study population consisted of household heads, community leaders, non-governmental organisation (NGO) representatives, CHOs, as well as officers from the NMCP and the GHS.

### Study design
We employed a concurrent triangulation mixed methods research design, involving participatory approaches within an implementation research framework. A desk review, survey, in-depth interviews, key informant interviews (KIIs), focus group discussions (FGDs) and participatory workshops (PWs) were organised concurrently, although at different times.

### Data collection
#### Desk review
A desk review was done to identify documents (ie, articles and reports) containing guidance and recommendations on effective strategies relating to mass LLIN distribution campaigns adopting community-based approaches in Ghana and elsewhere. This involved a comprehensive literature search and review to identify relevant published and grey literature. A desk review guide was used to collect appropriate data by initially presenting these in an extraction sheet that outlined potential barriers, enablers, lessons learnt and recommendations from similar interventions. Documents included scholarly journals, Ministry of Health documents and reports, GHS documents and reports, NMCP documents and reports, WHO/TDR documents, and reports, as well as documents and reports from the coalition of NGOs in health, an umbrella and coordinating body overseeing activities

of all registered NGOs and community-based organisations (CBOs) in the health sector of Ghana.

## Focus group discussions

FGDs (14) were organised to contextualise and explore the barriers to, and enablers of mass LLIN distribution campaigns in the context of community mobilisation, capacity building and SBCC. FGDs involved purposively selected household heads (4), caregivers of children under 5 (4), and CHOs (6).

## Key informant interviews

A semistructured interview guide was used to conduct 10 KIIs with purposively selected NMCP and GHS focal persons at the regional and district levels to assess LLIN campaign delivery processes.

## Baseline survey

A baseline survey (N=800) across the six districts was done to identify baseline parameters to be used for assessing the effectiveness of our cocreated intervention. These parameters included LLINs ownership, usage (regular, occasional and non-user), malaria morbidity among children under five and pregnant women. Data were collected using REDCap software on android tablets.

## Intervention cocreation

Findings from the Desk Review, FGDs, KIIs, and baseline surveys were synthesised, and grouped according to relevance. This was then distilled and formed the basis for developing a PW guide aimed at cocreating a LLIN campaign intervention involving various stakeholders (ie, investigators, NGO representatives, School Health Education Programme Coordinators, ANC nurses, Disease Control Officers, District Health Management Teams (DHMTs), CHOs, community leaders and opinion leaders). Six PWs were conducted employing the participatory learning in action technique, which is a practical, adaptive research strategy that enables diverse groups and individuals to learn, work and act together in a cooperative manner, to focus on issues of joint concern, identify challenges and generate positive responses in a collaborative and democratic manner.[15]

Findings from the PWs (figure 1) were further synthesised to cocreate our intervention.

## Data analysis

As part of the statistical analyses, we initially performed correlation and frequency analyses to identify and prioritise elements in LLIN campaign processes to inform the content of our PW guide. Also, thematic framework

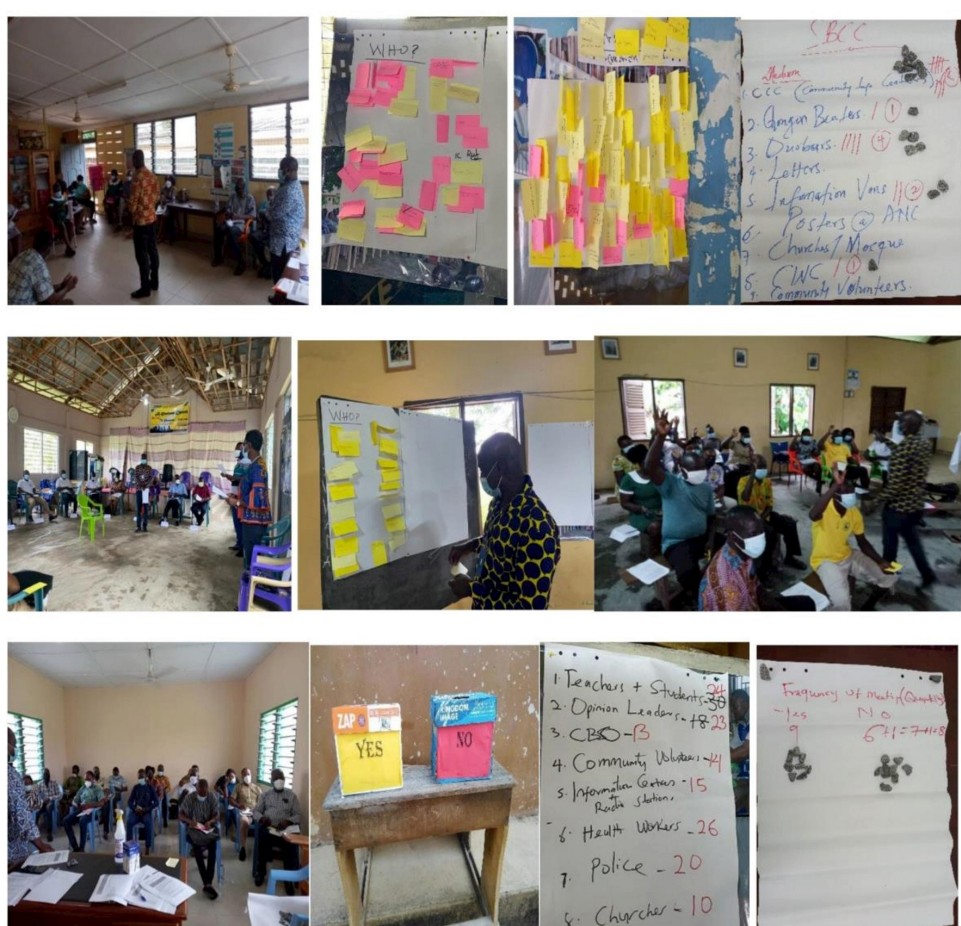

**Figure 1** Images of participatory workshops held in the communities.

Community Health Advocacy Team (CHAT) for LLIN Campaign Intervention

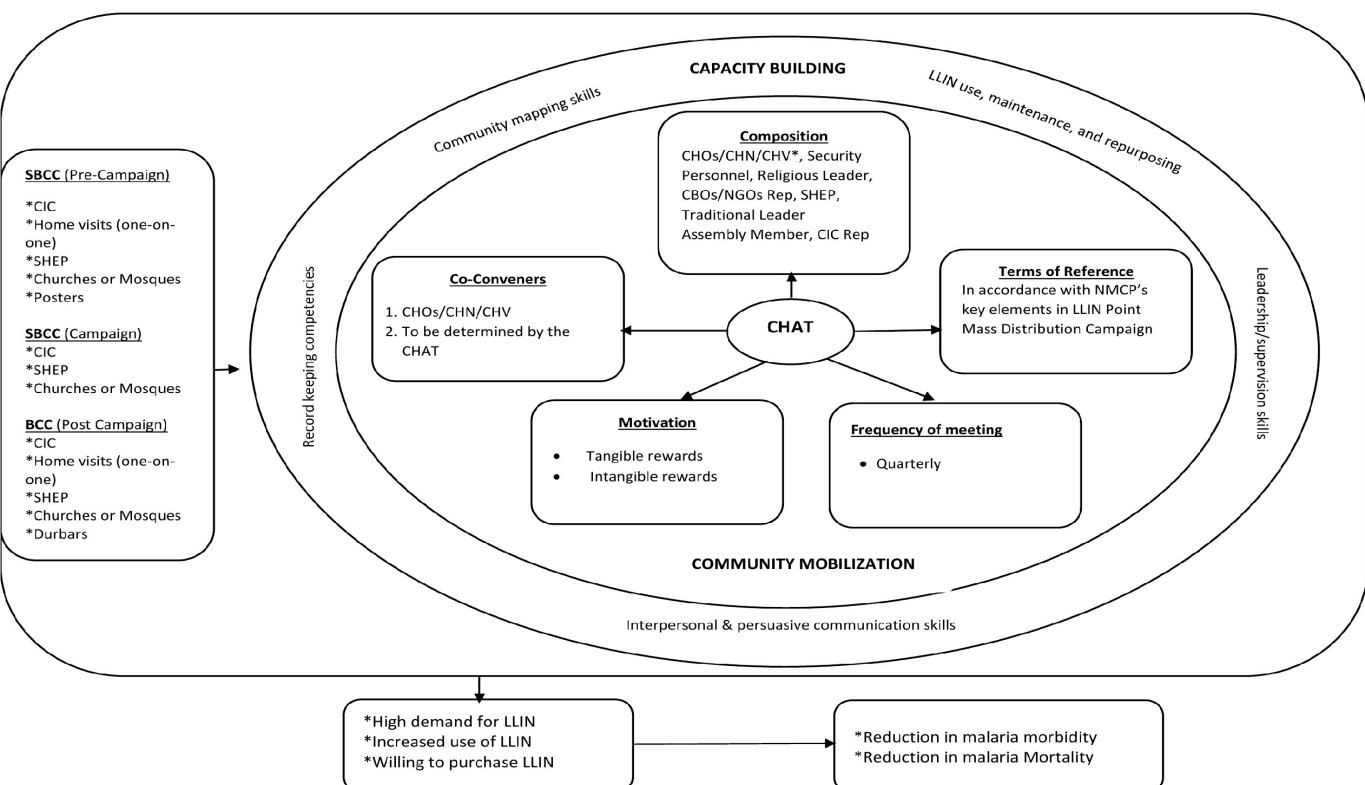

**Figure 2** Community health advocacy team (CHAT) for LLIN campaign intervention. CBO, community-based organisation; CHN, community health nurse; CHO, community health officer; CHV, community health volunteer; CIC, community information centre; LLIN, long-lasting insecticide net; NGO, non-governmental organisation; NMCP, National Malaria Control Programme; SBCC, social and behaviour change communication; SHEP, School Health Education Programme.

analysis was used to identify emerging themes to guide the establishment of CHAT, focusing on community mobilisation, capacity building and SBCC (figure 2). We used NCapture and query functions in the NVivo Program to display data and findings in the form of word clouds for easy visualisation.

## RESULTS
### Sociodemographic characteristics of participants
A total of 106 participants comprising 50 women and 56 men, aged between 20 and 77 years, were involved in the PWs. Participants were mainly NGO representatives, School Health Education Teachers, ANC nurses, Disease Control Officers, DHMTs, CHOs, Community leaders/Assemblymen, and opinion leaders (table 1). Participants for the PW were from Kukurantumi, Achiase and Apoli in the Eastern Region as well as HoKpeta, Tsito and Kpetoe from the Volta Region of Ghana.

### Community health advocacy team
Findings from this study suggested that the establishment of a CHAT will be very important in facilitating LLIN distribution campaigns within communities in Ghana (figure 2). CHAT will be made up of significant actors whose influences are recognised within communities. They will include CHOs, religious leaders, School Health Education Programme coordinators, assemblymen/women, community information officers, representatives from any of the security services, CBOs and traditional authorities. The anticipated role of CHAT would be three pronged. These are community/social mobilisation, capacity building and SBCC which will lead to improved use of LLIN. We found that the CHAT meetings are to be best convened quarterly, preferably by a CHO. Although it was emphasised that CHAT's efforts would not be compensated, there was a strong opinion among study participants that CHAT members be motivated tangibly or intangibly. For example, that they are given parcels of land where they could build (tangible) or be acknowledged publicly during community events (intangible).

### Community/social mobilisation
The NMCP in Ghana through its focal persons reach out to community members during LLIN distribution campaigns. These efforts could however be complemented by the CHAT since their positioning in the community make them readily available to support the NMCP. For CHAT to play this role creditably, its members must ensure the maintenance of NMCP's programming and its benefits over time.

**Table 1** Sociodemographic characteristics of participatory workshop participants

| Characteristic of participants | Number of participants |
|---|---|
| Community of residence | |
| Kukurantumi | 18 |
| Achiase | 20 |
| Apoli | 19 |
| HoKpeta | 17 |
| Tsito | 17 |
| Kpetoe | 15 |
| Total | 106 |
| Sex | |
| Female | 50 |
| Male | 56 |
| Total | 106 |
| Age | |
| <20 years | – |
| 20–29 years | 10 |
| 30–39 years | 61 |
| 40–49 years | 18 |
| 50+ years | 17 |
| Total | 106 |
| Educational level | |
| Primary | 1 |
| JHS/secondary/middle school | 17 |
| Tertiary | 88 |
| Total | 106 |
| Marital status | |
| Single | 28 |
| Married | 77 |
| Divorced/widowed/separated | 1 |
| Total | 106 |
| Length of stay in community | |
| <20 years | 84 |
| 20–29 years | 7 |
| 30–39 years | 4 |
| 40–49 years | 1 |
| 50+ years | 10 |
| Total | 106 |

JHS, Junior High School.

## Capacity building

Findings reveal that CHAT must be trained by the NMCP as part of their capacity building efforts. CHAT would therefore need to be trained along the themed capacity building areas (training, registration, SBCC, logistics, distribution and supervision) of the NMCP. In addition, CHAT members would be given skill enhancing strategies in leadership, communication and community mapping, as well as record keeping competencies.

## Social and behaviour change communication

The NMCP offers SBCC as part of the LLIN distribution campaign efforts. This study however revealed that the SBCC efforts could be strengthened if CHAT is actively involved at different stages of the campaign. For example, there could be SBCC activities prior to the campaign, during and after the campaign. Some of the channels identified include the use of posters, community information centres, home visits and in churches and mosques. This would ensure that communities are well sensitised before and after the campaign.

## DISCUSSION

This study used a participatory process[18 19] to develop a framework (campaign transitioning strategy) that could transition LLIN distribution into the CHPS structure, which is the primary healthcare system in Ghana. We actively involved various stakeholders to strengthen the development of an intervention that has the potential to systematically address real-world problems,[16] and to achieve sustainable outputs and impact[20 21] in malaria prevention and control. This would also help to achieve the global agenda of eliminating malaria.

From a practical point of view, this framework will shape the policy on LLINs distribution to ensure a continuous all year-round campaign on the regular use of LLINs. This would not only help address the challenges associated with periodic campaign rolled out during the PMD but ensure a reduction in the access to use gap. Another practical contribution of this framework would be the development of a guideline and training manuals on capacity building that support the transfer of CHAT to other communities. The use of local resources makes the intervention sustainable and easily integrated into the healthcare delivery system in Ghana. If adopted, the CHAT could become one of the flagship malaria prevention and control interventions which can contribute toward malaria elimination. The framework would also be suitable during public health emergencies that tend to disrupt facility-based service delivery.[22] Disruptions to malaria control programmes during public health emergencies have been linked to over 75 major resurgences arising from the 1930s through the 2000s.[23]

In summary, the present foundational study provides a framework for transitioning LLINs campaign into the primary healthcare system in LMIC settings. In doing so, it will add value to implementation science and practice concerning the scaling up and advancement of such strategies to address challenges in LLINs campaigns. The next step is for investigators to assess the acceptability and feasibility of the CHAT campaign transitioning intervention. We are mindful of an obvious limitation which will be to consider contextual issues (malaria prevalence, social and cultural) that may relate to particular communities.

**Contributors** All authors participated in designing the study with NYP and MC providing technical support. PA revised the final protocol. FG, EA and PT-NT collected and analysed the quantitative data while PD-G was in charge of qualitative data collection and analysis. AN carried out the desk review. RH revised and finalised the CHAT framework. All authors participated in the participatory workshops. All authors contributed to writing the manuscript and approved of the final draft. EA is responsible for the overall content as the guarantor.

**Funding** This work received financial support from the Health Campaign Effectiveness Programme, which is funded by the Bill & Melinda Gates Foundation at The Task Force for Global Health.

**Disclaimer** The authors alone are responsible for the views expressed in this article, and they do not necessarily represent the decisions or policies of PAHO or TDR. In any reproduction of this article there should not be any suggestion that PAHO or TDR endorse any specific organisation services or products.

**Competing interests** None declared.

**Patient and public involvement** Patients and/or the public were involved in the design, or conduct, or reporting, or dissemination plans of this research. Refer to the Methods section for further details.

**Patient consent for publication** Not applicable.

**Ethics approval** This study was approved by Ghana Health Service Ethics Review Committee Reference Number: GHS-ERC: 002/06/21. Participants gave informed consent to participate in the study before taking part.

**Provenance and peer review** Not commissioned; externally peer reviewed.

**Data availability statement** Data are available upon reasonable request.

**ORCID iDs**
Emmanuel Asampong http://orcid.org/0000-0002-1926-1118
Phyllis Dako-Gyeke http://orcid.org/0000-0002-4632-1833

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
