## [Reviewer comments · BMJ Open]

This paper was submitted to a another journal from BMJ but declined for publication following peer review. The authors addressed the reviewers' comments and submitted the revised paper to BMJ Open. The paper was subsequently accepted for publication at BMJ Open.

(This paper received two reviews from its previous journal but only one reviewer agreed to published their review.)

ARTICLE DETAILS

TITLE (PROVISIONAL)	Creating Interventions to Transition Long Lasting Insecticide Net Distribution in Ghana
AUTHORS	Glozah, Franklin; Asampong, Emmanuel; Tabong, Philip; Nwameme, Adanna; HORNUNVO, RUBY; Chandi, Margaret; Peparah, Nana; Adongo, Philip; Dako-Gyeke, Phyllis

VERSION 1 – REVIEW

REVIEWER	Lenore Manderson Wits University, Public Health
REVIEW RETURNED	27-Nov-2021

GENERAL COMMENTS	This article on interventions to upscale the LLIN distribution campaign in Ghana highlights the complexity and extent of the efforts needed to bring communities on board in order for any health campaign to be successful. The innovation discussed in this article focuses on integrating advocacy and community education on the use of LLIN into primary healthcare services, involving volunteers, to ensure that preventive methods (through the use of LLIN) is sustainable. This article will be of considerable interest to people working in the control of vector-borne diseases and community engagement, but a number of things could help improve it. Some of these are minor; I discuss all below. The title is very long and I really think that it could be shortened – for example – “Creating interventions to transition LLIN distribution in Ghana” is all that is needed. Page 1 of the manuscript, line 13, the sentence on the Ghana Malaria Survey needs to be better linked to the sentence that follows, when the authors return to the LLIN campaign. The point being made, I assume is that people have not taken up LLIN despite the distribution campaign Page 1, line 31, delete “behind such innovations” as this is clear from the remainder of the sentence and the previous sentence Page 1 line 57 – delete “bigger”, not needed Page 3, line11 – “lists” not “listing of” Page 3, line 12 “distributes LLINs” (not -- and distribution of LLINs) Page 3, line 14-15 – “paid for on a contractual basis.” Page 3, line 22... innovative social intervention which would transition Page 3, line 36 – since the focus is community, I suggest this first paragraph in Methods begins “A total of six communities, one per
--

	district across two regions in southern Ghana, participated in the study.” Page 3, line 40 --PMD campaigns are ongoing-- is there a reference for this please. Page 4, line 8: Organisation (NGO) representatives – NGO is an adjective here and so should be in the singular Page 4, Line 49 – what coalition of NGOs? Are there more than one – a brief discussion of this would be useful. Page 5, line 17: data were collected (NOT data was) Page 6 line 15 – spell out SBCC at first mention Page 6, line 17 – “query functions in the NVivo program” (in not on) Page 7, Line 9– 50 women and 56 men (noun not adjective) The number of people involved in this study is really impressive, including within government and among political leaders, traditional authorities, civil society, and at community and organisational levels. But I would like much more information about the engagement that brought all of these people on board and the length of time this took. This will be important and might inspire others to adopt this approach, but also, it makes the point that it is not as simple as a town hall meeting... Page 8, line 56 – tangibly or intangibly – what would an intangible motivation be? Again this section could be elaborated slightly. It is really critical to understand how volunteer are motivated to participate in such programs Page 9. line 54 – and in churches and mosques (or in collaboration of churches and mosques) Page 11, Discussion, paragraph 2. The authors write of this framework as the basis of shaping the policy on LLINs distribution: Are the authors suggesting that the framework has been adopted by the Malaria Control Program, or that this is a proposal that could be adopted to roll out LLINs?? What steps have been taken to encourage the uptake of this approach? What kind of discussions occurred during the study with the Ministry of Health to integrate community health workers into MCP activities, and to work in advocacy and support the use of LLINs at a community level? The authors also note: “another practical contribution of this framework would be the development of a guideline and training manual” – I am confused by “would be” – so this has not happened yet? Has a guideline been developed yet? Have training manuals been developed? Or is this work which the authors are proposing in order to implement this innovation? Page 11, line 37 – easily integrated – not integrable Page 11, line 48 - resurgences in the past – over what period of time? Page 11, line 53... LLINs campaign into the primary healthcare system... (Insert “the”). This again implies that the framework has been developed but that this is not been implemented. I suggest that you make clearer that the research being described in this paper is foundational work in order for this approach to be adopted. Page 12, final sentence: in what ways does malaria prevalence impact on transitioning the intervention? And what kind of social and cultural factors related to particular communities are relevant to this, and which might limit the impact of the intervention? Acknowledgement – delete quote marks; include reference to ethics approval also.
--	---

	References – the reference style is inconsistent and some data are missing – the authors need to work carefully through this to ensure correct bibliographic entries.
--	---

VERSION 1 – AUTHOR RESPONSE

Comment: The title is very long and I really think that it could be shortened – for example – “Creating interventions to transition LLIN distribution in Ghana” is all that is needed.

Response: Thank you very much. The title has been changed to the suggested title.

Comment: Page 1 of the manuscript, line 13, the sentence on the Ghana Malaria Survey needs to be better linked to the sentence that follows, when the authors return to the LLIN campaign. The point being made, I assume is that people have not taken up LLIN despite the distribution campaign

Response: Many thanks. The sentence has been appropriately linked with the introduction of this: “The inference therefore is that people have not taken up LLIN despite the distribution campaign”

Comment: Page 1, line 31, delete “behind such innovations” as this is clear from the remainder of the sentence and the previous sentence

Response: Thanks. “behind such innovations” has been deleted.

Comment: Page 1 line 57 – delete “bigger”, not needed

Response: Thank you. The word “bigger” has been deleted.

Comment: Page 3, line 11 – “lists” not “listing of”

Response: Many thanks. The word “lists” has been used instead of listing.

Comment: Page 3, line 12 “distributes LLINs” (not -- and distribution of LLINs)

Response: Many thanks again. That correction has been done.

Comment: Page 3, line 14-15 – “paid for on a contractual basis.”

Response: Thanks. This correction has been done to read: “paid for on a contractual basis.”

Comment: Page 3, line 22... innovative social intervention which would transition

Response: Thanks for the correction. It has been done as suggested.

Comment: Page 3, line 36 – since the focus is community, I suggest this first paragraph in Methods begins “A total of six communities, one per district across two regions in southern Ghana, participated in the study.”

Response: Thank you. The suggestion has been accepted and effected.

Comment: Page 3, line 40 --PMD campaigns are ongoing-- is there a reference for this please.

Response: Thanks. There is no reference for this as such. However, working in collaboration with the National Malaria Control Programme (NMCP) in this study, we had access to their Draft plan for the 2021 Long Lasting Insecticide Nets (LLINs) Point Mass Distribution (PMD) campaign.

Comment: Page 4, line 8: Organisation (NGO) representatives – NGO is an adjective here and so should be in the singular.

Response: Many thanks. NGO been corrected to reflect as singular.

Comment: Page 4, Line 49 – what coalition of NGOs? Are there more than one – a brief discussion of this would be useful.

Response: Thank you very much. Some brief description of the Coalition of NGOs has been provided in the text “umbrella and coordinating body of activities of all registered NGOs/Community Based Organisations (CBOs) in the health sector in the country”

Comment: Page 5, line 17: data were collected (NOT data was)

Response: Thanks. Correction is done.

Comment: Page 6 line 15 – spell out SBCC at first mention

Response: Thank you. SBCC is now first mentioned on Page 2 and spelt out.

Comment: Page 6, line 17 – “query functions in the NVivo program” (in not on)

Response: Many thanks. The correction has been done.

Comment: Page 7, Line 9– 50 women and 56 men (noun not adjective)

Response: Thanks. The noun “women” and “men” has been used.

Comment: The number of people involved in this study is really impressive, including within government and among political leaders, traditional authorities, civil society, and at community and organisational levels. But I would like much more information about the engagement that brought all these people on board and the length of time this took. This will be important and might inspire others to adopt this approach, but also, it makes the point that it is not as simple as a town hall meeting...

Response: Thanks for the comment. Indeed, this was a carefully thought through plan executed to get the number of people involved in this study. Providing all the information exceeds the number of words as required by the journal hence our tilt towards a summary of how we got the people. For example, Purposive sampling was used to select household heads for the FGDs. Upon arrival in the community, the study team met the Ghana Health Service (GHS) officials who facilitated meetings with local leaders. After gaining the support of the community leaders, the team assisted by community-based volunteers and an officer of the district health management team, recruited study participants.

Comment: Page 8, line 56 – tangibly or intangibly – what would an intangible motivation be? Again, this section could be elaborated slightly. It is really critical to understand how volunteer are motivated to participate in such programs

Response: Thanks. An example of tangible and intangible motivation as expressed by participants have been provided in the text.

Comment: Page 9. line 54 – and in churches and mosques (or in collaboration of churches and mosques)

Response: Thank you. “and in churches and mosques” has been used.

Comment: Page 11, Discussion, paragraph 2. The authors write of this framework as the basis of shaping the policy on LLINs distribution: Are the authors suggesting that the framework has been adopted by the Malaria Control Program, or that this is a proposal that could be adopted to roll out LLINs?

Response: Many thanks. We collaborated with the National Malaria Control Programme (NMCP) right from the development of the proposal and the execution of the study. This Framework is thus a proposal to the NMCP.

Comment: What steps have been taken to encourage the uptake of this approach? What kind of discussions occurred during the study with the Ministry of Health to integrate community health workers into MCP activities, and to work in advocacy and support the use of LLINs at a community level?

Response: Many thanks again. As mentioned earlier, NMCP officials have been part of the study team and are abreast with all the activities undertaken by the project team. They have also been part of all the visits to the study areas. Before, during and after the data collection, we have had periodic meetings with the Ghana Health Service through their Regional and District Health Directors.

Comment: The authors also note: “another practical contribution of this framework would be the development of a guideline and training manual” – I am confused by “would be” – so this has not happened yet? Has a guideline been developed yet? Have training manuals been developed? Or is this work which the authors are proposing in order to implement this innovation?

Response: Thanks. The Project Team has developed a Training Manual which was used in providing training to CHAT members a month ago.

Comment: Page 11, line 37 – easily integrated – not integrable

Response: Many thanks. The correction has been done.

Comment: Page 11, line 48 - resurgences in the past – over what period of time?

Response: Thank you. The period for these resurgences has been indicated in the text “from the 1930s through the 2000s”.

Comment: Page 11, line 53... LLINs campaign into the primary healthcare system... (Insert “the”). This again implies that the framework has been developed but that this is not been implemented. I suggest that you make clearer that the research being described in this paper is foundational work in order for this approach to be adopted.

Response: Thank you very much. “the” has been inserted. We have also introduced the word “foundational” as suggested (“In summary, the present foundational study provides a framework.....”)

Comment: Page 12, final sentence: in what ways does malaria prevalence impact on transitioning the intervention? And what kind of social and cultural factors related to particular communities are relevant to this, and which might limit the impact of the intervention?

Response: Thank you. It is expected that the roles to be played by the CHAT would improve LLIN use which would in turn reduce malaria prevalence. We also acknowledge that there are differences in socio-cultural practices which need to be taken into account in the roll-out of the intervention.

Comment: Acknowledgement – delete quote marks; include reference to ethics approval also.

Response: Thanks. The quote marks have been deleted; reference for ethics approval is included already on Page 6 “Ghana Health Service Ethics Review Committee (GHS-ERC: 002/06/21)

Comment: References – the reference style is inconsistent and some data are missing – the authors need to work carefully through this to ensure correct bibliographic entries.

Response: Thanks. We have added on new references and made corrections to the earlier ones.